# Evolution of the *WRKY66* Gene Family and Its Mutations Generated by the CRISPR/Cas9 System Increase the Sensitivity to Salt Stress in *Arabidopsis*

**DOI:** 10.3390/ijms24043071

**Published:** 2023-02-04

**Authors:** Youze Zhang, Peng Li, Yuqian Niu, Yuxin Zhang, Guosong Wen, Changling Zhao, Min Jiang

**Affiliations:** 1Ministry of Education Key Laboratory for Biodiversity Science and Ecological Engineering, Center for Evolutionary Biology, School of Life Sciences, Fudan University, Shanghai 200438, China; 2State Key Laboratory of Tea Plant Biology and Utilization, Anhui Agricultural University, Hefei 230036, China; 3Shanghai Key Laboratory of Plant Functional Genomics and Resources, Shanghai Chenshan Botanical Garden, Shanghai 201602, China; 4Research & Development Center for Heath Product, College of Agronomy and Biotechnology, Yunnan Agricultural University, Kunming 650201, China

**Keywords:** WRKY transcription factor, evolution, CRISPR/Cas9, salt stress, ABA, RNA-seq

## Abstract

Group Ⅲ WRKY transcription factors (TFs) play pivotal roles in responding to the diverse abiotic stress and secondary metabolism of plants. However, the evolution and function of *WRKY66* remains unclear. Here, *WRKY66* homologs were traced back to the origin of terrestrial plants and found to have been subjected to both motifs’ gain and loss, and purifying selection. A phylogenetic analysis showed that 145 *WRKY66* genes could be divided into three main clades (Clade A–C). The substitution rate tests indicated that the *WRKY66* lineage was significantly different from others. A sequence analysis displayed that the *WRKY66* homologs had conserved WRKY and C2HC motifs with higher proportions of crucial amino acid residues in the average abundance. The AtWRKY66 is a nuclear protein, salt- and ABA- inducible transcription activator. Simultaneously, under salt stress and ABA treatments, the superoxide dismutase (SOD), peroxidase (POD) and catalase (CAT) activities, as well as the seed germination rates of *Atwrky66*-knockdown plants generated by the clustered, regularly interspaced, short palindromic repeats/CRISPR-associated 9 (CRISPR/Cas9) system, were all lower than those of wild type (WT) plants, but the relative electrolyte leakage (REL) was higher, indicating the increased sensitivities of the knockdown plants to the salt stress and ABA treatments. Moreover, RNA-seq and qRT-PCR analyses revealed that several regulatory genes in the ABA-mediated signaling pathway involved in stress response of the knockdown plants were significantly regulated, being evidenced by the more moderate expressions of the genes. Therefore, the AtWRKY66 likely acts as a positive regulator in the salt stress response, which may be involved in an ABA-mediated signaling pathway.

## 1. Introduction

It’s well known that saline soil greatly influences seed germination, crop growth and productivity. In fact, salt stress affects the growth and development of plants throughout their life cycles, resulting in membrane damage, growth retardation, lipid peroxidation, ion balance disturbance, reactive oxygen species (ROS) accumulation and so on [1]. Electrolyte leakage (EL) and malondialdehyde (MDA) content are the important indicators of oxidative damage, and they increase under salt stress [2]. Usually, salt stress results in the increase of cytosolic free calcium concentration ([Ca^2+^]_cyt_). Hence, plants can respond to salt stress by activating a calcium-dependent protein kinase pathway, known as the salt overly sensitive (SOS) pathway [3]. Namely, the Ca^2+^-binding protein SOS3 (exports Na^+^ ions out of cells and is highly conserved in plants) [4] can physically interact with and activate the SOS2, thereby phosphorylates, and can activate the SOS1 Na^+^/H^+^ antiporter (increases Na^+^ efflux) [5,6]. In addition, salt stress can trigger the abscisic acid (ABA) signaling pathway by activating the sucrose non-fermenting 1-related protein kinases 2 (SnRK2s) activity [7,8]. However, there is also some evidence that the salt/osmotic stress-mediated SnRK2 activation also acts in an ABA-independent manner [9]. 

Transcription factors (TFs), such as WRKY, AP2 (APETALA2)/ERF (ethylene-responsive factor), MYB and NAC [no apical meristem (NAM), *A. thaliana* transcription activation factor (ATAF1/2) and cup-shaped cotyledon (CUC)], as well as other gene expression-regulating components, play critical and unique roles in the plant adaptation to abiotic stresses [10]. Particularly, the WRKY is one of the largest TF families in plants with a conserved domain of 60 amino acid residues which contain a DNA-binding domain (WRKY domain) with a highly conserved WRKYGQK heptapeptide sequence and a C_2_H_2_ or C_2_HC zinc finger motif [11], and the consensus W-box (TTGACT/C) *cis*-elements of the target gene promoters are the core binding motif of the WRKY [12]. WRKY members fall into three groups (I, Ⅱ and Ⅲ) and multiple subgroups (e.g., Ⅱa, Ⅱb, etc.) based on their phylogenetic relationships [13]. Group Ⅲ proteins have one WRKY motif along with a C2HC zinc finger motif [14]. Under abiotic stresses, such as drought, wounding, high and low temperatures, high salt and heavy metal contamination, multiple *WRKY* genes are strongly expressed [15,16]. The SWEET POTATO FACTOR1 (SPF1), a first-identified WRKY, negatively regulated the expression of β-amylase and sporamin genes [17], while *ItfWRKY70* improved the drought tolerance of *Ipomoea trifida* [18]. Under drought and cold stresses, the seed germination and root growth both increased in the *Arabidopsis* overexpression of *PoWRKY1* (*Polygonatum odoratum*) [19]. *WRKY70* and *WRKY54* conferred osmotic stress tolerance by negatively regulating stomatal closure in *Arabidopsis* [20]. In addition, *WRKYs* can be induced by a variety of signal substances including salicylic acid (SA), ethylene (ET), jasmonic acid (JA), gibberellin (GA) and ABA [21,22,23]. For instance, AtWRKY57 functioned as a repressor in the JA-induced leaf senescence process [24]. Similarly, the overexpression of *AtWRKY75* led to the precocious leaf senescence induced by SA- and ROS-mediated signaling [25]. Likewise, the overexpression of *WRKY6* displayed ABA-hypersensitive phenotypes during the early seedling development and seed germination in *Arabidopsis* [26]. However, the functions of the *WRKY66* in response to abiotic stresses and plant hormones have not been directly clarified by using its overexpression plants or loss-of-function mutants.

It is well-known that the loss-of-function mutant is the crucial component in gene function studies and, in recent years, genome editing has been an important dynamic technique to acquire the mutant. In gene editing, several sequence-specific nucleases, e.g., zinc-finger nucleases (ZFNs), transcription activator-like effector nucleases (TALENs), and clustered, regularly interspaced, short palindromic repeats (CRISPR)-associated protein 9 (Cas9) systems have been successfully applied in plant gene-editing [27]. Among them, the CRISPR/Cas9 system has been widely employed to edit plant genomes due to its high-efficiency, stabilization and simplicity [28]; this system enables the generation of specific double-stranded DNA breaks (DSBs) at a site complementary to the guide RNA (gRNA) sequence 2–4 bp upstream of the protospacer adjacent motif (PAM) sequence [29], which allows this system to efficiently implement point mutations including the nucleotide insertions or deletions of target genes. Thus, the enzyme site on the gRNA sequence upstream of PAM is helpful for the next identification of positive plants [30]. To date, the effective CRISPR/Cas9 system has been increasingly applied in various plants, such as rice [31], sorghum [32], wheat [33], bananas [31,34], tomatoes [35] and so on.

Here, we investigated the *WRKY66* homologs using bioinformatics methods and analyzed the taxonomic distribution, structural diversification following duplication, molecular evolution and phylogenetic relationships throughout plant lineages. To better understand the function of the *Arabidopsis WRKY66* gene (*AtWRKY66*), expression patterns were surveyed under salt stress and ABA treatments. Moreover, the subcellular localization analysis and transcription activation assay of the *AtWRKY66* were also performed and observed. Furthermore, the CRISPR lines of the *AtWRKY66* generated by the CRISPR/Cas9 system were performed, and analyzed the phenotypes under salt stress and ABA treatments including the relative electrolyte leakage (REL), seed germination rate and the activities of superoxide dismutase (SOD), peroxidase (POD) and catalase (CAT). Finally, RNA-seq and quantitative real-time PCR (qRT-PCR) analyses of *WRKY66* CRISPR lines under salt stress and ABA treatments were performed. Overall, these results provided a new perspective on the origin and evolutionary history, and laid a foundation for further functional research, of plant *WRKY66s*.

## 2. Results

### 2.1. Phylogenetic and Molecular Evolutionary Characteristics of the WRKY66 Homologs

A total of 145 *WRKY66* homologs were identified from 28 plant genomes (Appendix A) and recruited to reconstruct a phylogenetic tree. It is clear that the number of main *WRKY66* homologs and the relationships between these are from the tree (Figure 1A). The phylogenetic tree showed that the *WRKY66* homologs split into three main clades, i.e., Clade A, B (*WRKY30* lineage) and C (*WRKY66* lineage) (Figure 1A), and could be traced back to the origin of terrestrial plants (Figure 1B). Obviously, the last common ancestor of the extant terrestrial plants possessed at least one *WRKY66* homologous gene, but the *WRKY66* orthologs were only present in angiosperm plants originating via the duplications and subsequent diversification of an ancestral gymnosperm gene (Figure 1B). Furthermore, the *WRKY66* homologs held a highly conserved WRKY motif and a C2HC type zinc finger motif. Remarkably, four new highly conserved motifs (motif 4, 5, 6 and 11) were also identified (Appendix A). The comparisons between the *WRKY66* ortholog proteins and others revealed that several motifs were lost following the gene duplication (Figure 1C and Appendix A). Additionally, to further identify the consensus sequences’ features among the phylogenetic groups, the WebLogo 3 online tool was used to analyze the sequence logo motifs of each group to elucidate the sequence features of functional domains, e.g., the detailed information of the WRKY motifs (Appendix A). Except for the typical WRKYGQK motif, some motif variants including WRKYGKK (*SlyWRKY30C*), WSKYEQK (*OsaWRKY30F*) and WKRYGEK (*OsaWRKY30s*) could be also found in plants. The WRKY66 homolog proteins held group-specific motif variants, while they displayed conservativeness in the C2HC motif (Appendix A). 

Moreover, the isoelectric point (pI) and molecular weight (Mw) of different WRKY66 homolog proteins were also surveyed. The size of pI ranged from 4.32 (GbiWRKY30_66B) to 10.50 (BdiWRKY30B), and the size of Mw of WRKY66 homolog proteins ranged from 12.971 (PsyWRKY30_66A) to 79.521 (ApuWRKY30_66) kDa (Appendix A), which were consistent with the amino acid residue numbers ranging from 119 (PsyWRKY30_66A) to 761 (ApuWRKY30_66) amino acids (Appendix A). Interestingly, those of all the WRKY66 homologs in each group varied from acidic to alkaline. Furthermore, the average amino acid composition of the WRKY66s ranged from 0.97 (tryptophan) to 11.76 (serine) (Appendix A). The average abundance of the most important amino acids such as tryptophan (W), arginine (R), lysine (K), tyrosine (Y), glycine (G) and glutamine (Q) (WRKYGQK) which is necessary and suffice for DNA binding were 0.97, 5.07, 5.03, 2.96, 6.32 and 5.25, respectively; those of the cysteine (C) and histidine (H) in the C2HC motif were 2.01 and 3.10, respectively; and, furthermore, those of the hydrophobic amino acids were relatively higher than those of other amino acids, such as alanine (A; 6.42), leucine (L; 7.01), proline (P; 6.36) and valine (V; 4.50) (Appendix A).

Particularly, the genetic distance between the Clade A and Ancient *WRKY66s* was the lowest, i.e., 0.549 (Table 1), which was consistent with their evolutionary grouping (Figure 1A), whereas that between the Clade C and Ancient *WRKY66s* was the largest, indicating the least sequence similarity between them. Interestingly, the combinations of the Clade C with other *WRKY66* groups held higher genetic distances than those of other counterparts (Table 1), suggesting the higher sequence divergences of the Clade C WRKY66s and other counterparts. In addition, the overall mean distance of the *WRKY66s* was 0.723 (standard error 0.125). Moreover, the molecular evolution analysis displayed that the Clade C *WRKY66s* had the lowest Ka and ώ values (ώ = 0.4776), reflecting the strong purifying selection during the evolution process (Table 2). Similarly, substitution rate tests also indicated that all clades had the ώo values of lower than 1, which again implied the purifying selection. Further analysis found that the foreground value of the *WRKY66* lineage (ώf = 0.417126) was bigger than that of other *WRKY66* homologs (ώb = 0.353674) under the two-ratio model, revealing that the *WRKY66* lineage had been subject to a relaxed purifying selection (Table 3). Furthermore, there was the significant difference of the substitution rate between the *WRKY66* lineage and others (*p* < 0.001) (Table 3). Additionally, the mean ώ values of the Clade B and Ancient *WRKY66s* were both greater than 1, indicating that the genes also underwent a positive selection during the evolution (Table 2). Thus, the purifying selection was the most obvious characteristic of the molecular evolution of the *WRKY66s*.

### 2.2. Bioinformatics Characteristics and Prediction of the AtWRKY66 Gene

The *AtWRKY66* gene encodes a 236 aa protein with a typical WRKY motif and a C2HC type zinc finger motif, and it belonged to group Ⅲ of the *WRKY* family [36] (Figure 2A). The multiple sequence alignment showed that the amino acid sequence of the *AtWRKY66* was highly homologous to the amino acid sequence of the WRKY66s of *Arabidopsis*, *Oryza sativa*, *Zea mays*, *Vitis vinifera*, *Solanum lycopersicum*, *Glycine max*, *Fragaria vesca* and *Populus trichocarpa* (Figure 2A), meaning that the *WRKY66s* were exceedingly conserved in plants. Phylogenetic analysis indicated that the *AtWRKY66* belongs to group Ⅲ of the *WRKY* family with 14 members [37] (Figure 2B). Moreover, in the phylogenetic tree constructed on the AtWRKY66 and the group Ⅲ members of the other *Arabidopsis* WRKY, the *AtWRKY66* was identified to be closely related to *AtWRKY63* (Figure 2B). 

Based on the fact that most WRKYs in Group Ⅲ were involved in the stress response, it was preliminarily revealed by the *cis*-element analysis that the *AtWRKY66* might participate in the hormone-mediating signaling pathway in plants, which was mainly because the promoter region (389 bp upstream of the start codon) of the *AtWRKY66* gene contained some distinctive stress response-related elements, e.g., one element involved in the ABA responsiveness (ABRE), two elements involved in the Me-JA-responsiveness (CGTCA- and TGACG- motif), two elements involved in the GA-responsiveness (GARE-motif) and so on (Figure 2C and Appendix A).

### 2.3. The AtWRKY66 Is a Salt- and ABA- Inducible Transcription Activator

To investigate the possible function of the *AtWRKY66* in the abiotic stress response, we analyzed its expression patterns under salt stress and ABA treatments. qRT-PCR showed that the expression level of the *AtWRKY66* increased very significantly within 12 h after the NaCl treatment and peaked at 48 h with approximately 90 folds higher than that of the control (Figure 3A). Likewise, in the ABA treatment, the expression levels of the *AtWRKY66* increased dramatically within 3 h, peaked at 12 h with about 16 folds higher than that of the control, and recovered gradually at relatively high and striking levels (Figure 3B). Thus, in *Arabidopsis*, the *AtWRKY66* might be stress-responsive.

To further clarify the subcellular localization of the *AtWRKY66*, the transient transformation assay was performed in *Nicotiana benthamiana* leaves. Microscopic observation of the leaves showed that the AtWRKY66-GFP fluorescence signal distributed uniformly in the nuclei, whereas the GFP signal alone distributed throughout the entire cell without a specific compartment (Figure 3C), indicating that the AtWRKY66 might be a nucleus-localized protein. On the other hand, to identify the transcription activity of the *AtWRKY66*, we also constructed the *AtWRKY66* into pGBKT7 to produce the BD-*WRKY66* which was then transformed into yeast cells. The results displayed that the yeast strain carrying the BD-*WRKY66* grew normally, exhibited β-galactosidase activity and turned blue on SD/-Trp-His-Ade/X-α-gal media, while the transformants only containing BD could not grow (Figure 3D), demonstrating that the *AtWRKY66* held a transcription activation capacity.

### 2.4. Generation and Identification of the AtWRKY66 Transgenic CRISPR Lines

To further investigate the function of the *AtWRKY66* in *A. thaliana*, the knockdown mutants (*wrky66*) were generated by using the CRISPR/Cas9 system. Firstly, the best target site (TS) of the *AtWRKY66* was screened according to the design rule of the single guide RNA (sgRNA). The sgRNA contained an *NcoI* restriction enzyme site which facilitated the subsequent identification of the positive mutants. The TS (TCCCAAGAAATTACCATGG) was at the 3rd exon which corresponded to the 541–559 bp region in the coding sequence of the *AtWRKY66* (Figure 4A), and the TS sequence was located at the 180–188 aa section of the AtWRKY66 which resided in the C-terminal outside of the WRKY domain (Figure 4B). Then, after the constructs (*pYAO*:hSpCas9-*AtWRKY66*-sgRNA) were generated, the *Agrobacterium*-mediated transformation was employed to produce the *AtWRKY66* CRISPR lines, and the independent transgenic plants were detected by the PCR sequencing of the genomic DNA (gDNA). These results displayed that six independent transgenic plants of the T1 generation exhibited a single band of 1082 bp, while eight lines were partially cut off by the enzyme (Appendix A). In the same way, to eliminate the multi-allelic patterns and further verify the mutation situation, all seedlings of the T2 transgenic plants were surveyed with the method mentioned above. The results revealed that seven independent transgenic plants including the *wrky66-1*, *-4*, *-6*, *-8*, *-12*, *-14* and *-20* lines displayed the single band of 1082 bp, while the line *wrky66-7* was partially cleaved by the enzyme (Figure 4C). In fact, the independent lines had several types of insertion or deletion. For example, the *wrky66-1* and *-12* harbored a 1 nt deletion in the target region, while in *wrky66-4* and *-6*, an adenine (A) or guanine (G) insertion occurred, respectively, and the *wrky66-8* showed a 10 nt deletion in the target region, while a 13 nt deletion occurred in the *wrky66-14* (Figure 4D). More importantly, the single peak and band indicated that the lines were homozygous (Figure 4C,E). Moreover, qRT-PCR confirmed that the expression levels of the *AtWRKY66* in the *wrky66-1*, *-4*, *-6* and *-12* lines did not decrease, obviously in comparison with that of the wild type (WT), whereas, it was significantly decreased in the *wrky66-8*, -*14* and -*20* lines (Figure 4F), which might be caused by the nonsense-mediated mRNA decay (NMD) pathway mRNA decay initiated by pre-mature stop codons in the *AtWRKY66* mRNA. Therefore, the *AtWRKY66* could effectively obtain the loss-of-function modifications via the CRISPR/Cas9 system. 

### 2.5. The Decrease of the Salinity Stress Tolerance and the Increase of the ABA Sensitivity Resulting from the Loss-of-Function of the AtWRKY66

To determine the effects of the *AtWRKY66* on the responses to salt stress and ABA signaling, based on the mutation situations and expression levels in CRISPR lines, the *wrky66*-*8*, -*14* and -*20* transgenic plants were selected for further phenotypic, physiobiochemical and developmental analyses under the NaCl or ABA treatment. Firstly, there were significant growth differences between the mutant plants and the WT plants, especially with respect to the root length (Figure 5A,B). Secondly, the SOD, POD and CAT activities in the *Atwrky66* knockdown plants were all significantly lower than those in the WT plants under NaCl and ABA treatments (Figure 5C–E), suggesting that, as a TF, the *AtWRKY66* could positively regulate the transcriptions of the three enzyme genes and the enzyme activities. Concurrently, the *Atwrky66* knockdown plants displayed higher REL after the salt or ABA treatment (Figure 5F), indicating that these mutants were subject to more serious oxidative damage, and the loss-of-function of the *AtWRKY66* might weaken the plant tolerance to salt stress and ABA treatments through the cellular oxidative damage. Finally, the germination rate in the knockdown plants was lower (Figure 6A), e.g., after the NaCl treatment, the germination rate of the three knockdown plants was about 53.3%, 48% and 47%, respectively, while the rate of the WT plants was about 81.7% (Figure 6B). As a result, under the NaCl or ABA treatment, the survival rates of the knockdown plants were lower than those of the WT plants. 

### 2.6. The AtWRKY66 Affected the Expression Changes of the Salt Stress- and ABA-Related Genes

To investigate the molecular mechanisms of the *AtWRKY66* in response to salt stress or ABA treatment, the total RNAs extracted from the WT and *wrky66-14* knockdown plants under the normal and NaCl or ABA treatment were used for RNA-seq analysis. The results showed that the samples treated by the same condition were highly correlated on the sample–sample correlation heatmap (Figure 7A). By and large, 8867 differentially expressed genes (DEGs) were detected in the four experimental groups before and after the NaCl treatment (MUCK vs WTCK, WTNa vs WTCK, MUNa vs WTNa and MUNa vs MUCK, where ‘CK’ indicated the control, ‘Na’ indicated the salt stress exposure) (Figure 7B and Appendix A), and 6689 DEGs held significantly different transcript abundance before and after the ABA treatment (MUCK vs WTCK, WTABA vs WTCK, MUABA vs WTABA and MUABA vs MUCK, where ‘ABA’ indicated the ABA exposure) (Figure 7C and Appendix A). Moreover, the GO enrichment analysis exhibited that these DEGs were mainly involved in stress-responding (Appendix A). Among the DEGs, there were several salt stress- and ABA- related instances (Appendix A). Some DEGs showed the same expression trends between the WT and knockdown plants after the NaCl or ABA treatment. For example, the relative expression levels of the salt stress-related genes, including the *AKR4C8* (Aldo-keto reductase family 4 members C8), *ANN1* (annexin 1), *BGLU24* (β-glucosidase 24), *GSTU17* (Glutathione S-transferase U17), *SIGE* (sigma factor E) and *EGR1* (Clade E Growth-Regulating 1), were upregulated by the NaCl treatment, which was consistent with those treated by the ABA treatment (Table 4). Particularly, the expression levels of the *BGLU24* reached 12.04-fold in the WT plants, but only 3.60-fold in the knockdown plants, whereas the expression levels of the *AKR4C8*, *ANN1*, *GSTU17*, *SIGE* and *EGR1* in the WT plants decreased slightly in comparison with those of the mutants. Likewise, the expression levels of some ABA signaling-related genes, including the *ABA1* (ABA DEFICIENT 1), *MYB102* and *SnRK2.6*, also increased following the same trends in the WT and knockdown plants after the treatments (Table 4). Furthermore, the log_2_FC value of the expression level of the *MYB102* was 10.26 in the WT plants, but only 9.17 in the knockdown ones after the ABA treatment, but, under the NaCl treatments, the level difference was not obvious. In the meantime, the expression levels of particular salt stress- and ABA- related genes, including the *ALDH7B4* (aldehyde dehydrogenase 7B4), *CHIC* (class V chitinase), *NAC2* and *RD20* (RESPONSIVE TO DESSICATION 20), were also observed in the WT and knockdown plants after the two treatments (Table 4). For example, after the NaCl treatment, the expression level of the *RD20* drastically increased, reaching 24.05-fold in the WT plants, but only 9.16-fold in the mutants. Similarly, after the ABA treatment, the log_2_FC value of the expression level of the *RD20* was 9.40 and 8.58 in the WT plants and mutants, respectively.

Based on the KEGG pathways, we further constructed a schematic diagram of the activated ABA-mediated signaling pathways associated with salt stress (Figure 7D and Appendix A). The genes including the *PYRABACTIN RESISTANCE-LIKE* (*PYR*/*PYLs*), *protein phosphatases type 2C* (*PP2Cs*), *SnRK2s* and *MAPK* were significantly up- or down- regulated under the salt and ABA treatments. For instance, most of the class A *PP2Cs* were up-regulated under both treatments, while most *PYR*/*PYLs* of the ABA receptor were down-regulated in the ABA induction (Figure 7E and Appendix A). Similarly, the expression levels of the *CAT* and *SnRK2s* were also obviously down-regulated after the salt or ABA treatment (Figure 7E and Appendix A), which was consistent with the fast and transient activation of the *SnRK2*-related signaling and the conclusion that, in the stress response of plants, the signaling usually occurs in the early stages (minute level) and is attenuated in the later ones. Interestingly, in comparison with the WT plants, the mutants showed relatively small expression changes under both conditions, indicating that the mutation of the *WRKY66* might affect the plant’s response to the salt and ABA to some extent.

It should be noted that, in the qRT-PCR-mediated accuracy-validating of the RNA-seq and analysis, the expression patterns of the ten genes tested were consistent with the RNA-seq results, being evidenced by the correlation coefficient (r) values of higher than 0.9 (Appendix A). For instance, the r values of the *BGLU24* under the NaCl and ABA treatments were 0.993 and 0.983, respectively. Therefore, the *AtWRKY66*-caused changes in the expression levels of the ABA- and stress- related genes might be conducive to the increases of the salt stress tolerance and the ABA sensitivity of the *AtWRKY66* knockdown plants.

## 3. Discussions

### 3.1. Proposed Evolutionary History and Gene Duplication of the WRKY66 Homologs in Land Plants

Gene phylogeny reconstruction displayed that the *WRKY66* homologs had experienced duplication–divergence events in evolutionary history (Figure 1A). The ancestral genes of the *WRKY66* homologs had been duplicated in angiosperm genomes, prior to the radiation of extent flowering plants. Strictly speaking, the *WRKY66* orthologs were angiosperm-specific because they were all derived from the angiosperm-specific duplication events (Figure 1B), similar to the divergence pattern of the *PIF*, *BEL1* and *ANT* [38,39]. Then, the whole genome duplication giving rise to modern angiosperms [40], together with additional tandem duplications [41], thus created the genetic substrate for the structural and functional diversification of genes. Subsequent diversification of gene structure and expression patterns was observed in the paralogs of the angiosperm genomes and the homologs of gymnosperm (co-orthologs) that preceded those duplications. The *WRKY66* homologs showed structural variations in their coding regions. Of them, the *WRKY66* orthologs lost motifs in comparison with their paralogs, while the *WRKY30* lineages had undergone both a gain and loss of motifs (Figure 1C). In addition, the *TcWRKY20* (co-orthologs of the *WRKY66*) belonged to Group Ⅲ WRKYs and was up-regulated by Me-JA treatment and down-regulated by SA treatment, which acted on the taxol synthesis in *Taxus chinensis* [42]. Similarly, the *WRKY66* homologs, *CrWRKY1* and *AaWRKY1*, could positively regulate the synthesis of the vinblastine in *Catharanthus roseus* and artemisinin in *Artemisia annua*, respectively [43,44]. Therefore, even though, in different species, different secondary metabolites were formed, the *WRKY66* homologues, which positively regulated the secondary metabolite biosynthesis, were conserved. After duplication, relaxed constraints allowed paralogous genes to diverge through mutations or exon shuffling to gain or lose motifs/domains, affecting alternative splicing, protein–protein interaction and post-translational modification and, thereby, limiting in coding changes of the TFs in specific spatiotemporal contexts [45]. In this study, all ώo values were less than 1 and exhibited the purifying selection in the evolution of the *WRKY66* (Table 3), and substitution rate tests displayed the significant rate difference between the *WRKY66* lineage and others (Table 3), indicating that the *WRKY66* orthologs might occur through subfunctionalization or neofunctionalization.

### 3.2. Generation of the WRKY Knockdown Plants by Using the CRISPR/Cas9 System

In gene function research, the traditional methods to obtain mutants, such as natural mutations, physical or chemical mutations or random insertions into T-DNA, to determine gene function have many obvious limitations [46]. However, the *WRKY* knockouts generated by using genome editing technologies including the CRISPR/Cas9 system can efficiently, accurately and easily acquire the mutants with loss-of-function. For example, the knockout mutant of rice, *OsWRKY21* [47], the double knockout mutants of rice, *OsWRKY36* and *OsWRKY102* [48], the knockout lines, *BnWRKY47*, *BnWRKY11* and *BnWRKY70*, of *Brassica napus*, *WRKY* mutations in *N. benthamiana* and so on had been obtained by using the CRISPR/Cas9 system [49,50,51]. Recently, the double mutants, *AtWRKY3* and *AtWRKY4*, have also been acquired by the CRISPR/Cas9 system with their conserved sequence as sgRNA to direct Cas9 to the target site they were able to achieve [30]. Importantly, the selected gRNA sequences contain the restriction enzyme site (*NdeI*) in the upstream of PAM [30], which facilitated the subsequent identification of positive plants. Likewise, the selected gRNA sequences also comprise the restriction enzyme site (*NcoI*) in the present study. As a result, the *WRKY66* knockdown in *Arabidopsis* plants was successfully generated with the forms of insertions and/or deletions (indels) in the target sequences by using the CRISPR/Cas9 system (Figure 4). Furthermore, the transcriptional level of the *AtWRKY66* was confirmed to apparently decrease in the knockdown lines (Figure 4F).

### 3.3. Candidate Genes Involved in Response to High Salinity and ABA Treatments

The *AtWRKY66* exhibited a transcription activation function (Figure 3D), implying that it might function as a positive regulator of gene expression. Phenotypic analyses demonstrated that the mutations of the *AtWRKY66* reduced the plant tolerance to salt stress and increased the plant sensitivity to ABA, which was supported by the changes of the REL and the SOD, POD and CAT activities (Figure 5), as well as the seed germination rates (Figure 6). Salinity results in oxidative damage [3] that is generally indicated by the REL levels. Both drought and salt stresses increase the REL in plant tissues [52]. In addition, the ROS causes the accumulation of ROS-scavenging enzymes, such as SOD, POD and CAT [53]. In this study, the expression levels of the *CAT* significantly increased under the salt or ABA treatment to regulate the ROS homeostasis (Figure 7E). The *wrky66* knockdown plants also displayed higher REL and lower SOD, POD and CAT activities than those of the WT seedlings, indicating that the *AtWRKY66* might maintain the cell membrane integrity by modulating the cellular ROS levels under salt stress and ABA treatments (Figure 5).

We also employed the RNA-seq and analysis to elucidate the molecular mechanism of the reduced salt tolerance and increased ABA sensitivity in the *Atwrky66* knockdown plants. These results revealed that, in the knockdown plants, many salt stress- and ABA signaling- related genes were regulated by the NaCl or ABA treatments compared with the WT plants (Table 4). For example, after NaCl or ABA treatment, the expression level increase of the *BGLU24*, a member of the glycoside hydrolase family, which was reported to respond to salt tolerance, in the WT plants was more obvious than that in the mutants. Similarly, the expression levels of another five salt-responsive genes, *AKR4C8*, *ANN1*, *GSTU17*, *SIGE* and *EGR1* [54,55,56,57,58], also increased with the same trends. The stress-responsive gene *LEA3* was up-regulated in the two *CgZFP1* transgenic lines that conferred the salinity and drought tolerance under drought stress conditions [59], but its expression was down-regulated in the *Atwrky66* knockdown plants under the NaCl or ABA treatment (Table 4). Moreover, the knockdown plants also showed more sensitivity to salt stress and ABA treatments during the seedling stage (Figure 5A,B). Thus, the *AtWRKY66* might serve as a positive regulator to influence the expression levels of the stress-related genes in response to salt stress. 

Salt stress can also trigger the ABA-dependent signaling pathway in plants [6]. Likewise, in the current presented study, the *AtWRKY66* expression was strongly induced by the ABA treatment and salt stress (Figure 3A,B). The ABA signaling-related genes, the SnRK2 protein kinase gene (*SnRK2.6*) and group A PP2C-type phosphatase gene, constituting the core ABA pathway [60], were significantly up-regulated both in the mutant and WT plants (Table 4 and Figure 7E). Likewise, another three ABA signaling-related genes, *ABA1*, *MYB102* and *PLDDELTA* (phospholipase D delta) [61,62,63], were expressed as significantly high both in the mutant and WT plants under the two conditions. Interestingly, some genes, including *ALDH7B4*, *CHIC*, *NAC2* and *RD20* [64,65,66,67], were not only involved in response to salt stress, but also related to ABA signaling. For instance, under the two conditions, the expression levels of the *NAC2* which was associated with salt stress response and lateral root development through the ABA signaling pathway [64], increased both in the *wrky66* and WT plants. More importantly, an ABA signal transduction pathway involving the PYR/PYL ABA receptor and the phosphatase/kinase enzyme pairs, PP2Cs and SnRK2s, was activated under salt stress (Figure 7). However, the expression of the *PYR*/*PYLs* was down-regulated and that of *PP2Cs* was up-regulated under salt and ABA treatments (Figure 7E), which might reflect the characteristics of ABA. In fact, different ABA concentrations might lead to synergistic or antagonistic responses between ABA and other signals in response to stress to ensure moderate stress tolerance and be conducive to productive crop growth and development [68]. The outcome of the ABA signaling was the activation of a series of downstream salt- or ABA- related genes including those mentioned above by TFs under the control of the SnRK2s. The expression changes of the salt stress- and ABA- related genes were more intense in the WT plants than those in the mutants after the NaCl and ABA treatments. Consequently, the *AtWRKY66* played a positive regulatory role in mediating the salt tolerance through an ABA-dependent signaling pathway. In fact, the WRKY participates in an abiotic stress response and acts in an activator or repressor role in the ABA signaling [69]. For example, the overexpression of the *ItfWRKY70* conferred drought tolerance by modulating the stress-responsive genes via the ABA signaling pathway [18], and, similarly, the *WRKY114* and *WRKY17* in maize played negative roles in response to salt stress through the ABA signaling pathway [70,71]. 

## 4. Materials and Methods

### 4.1. Plant Materials, Growth Conditions and Treatments

The WT of the *A. thaliana* ecotype, Columbia-0 (Col-0), was used for the plant transformation. The seedlings were transferred into a soil mix (clay: soil = 3:1) and grown in an environment of 22 °C and 70% humidity with a 16 h-light/8 h-dark photoperiod. For the expression profile analyses, *Arabidopsis* seeds were surface sterilized and then placed in the dark at 4 °C for 2 days before sowing on a 1/2 Murashige and Skoog (MS) [72] solid medium (PhytoTechnology, Lenexa, KS, USA). The plates were then incubated under a 16 h-light/8 h-dark light cycle at a constant temperature of 22 °C. The 10-day-old seedlings were then transferred and treated with 100 mM NaCl (Sangon Biotech, Shanghai, China) or 10 µM ABA (Sangon Biotech, Shanghai, China) in MS liquid medium for 3 h, 6 h, 12 h, 24 h and 48 h, respectively. Samples were collected on the corresponding time points after the treatments and rapidly frozen in liquid nitrogen for further RNA extraction. Each sample contained at least 10 seedlings. Likewise, for the expression analysis of the *AtWRKY66* in the CRISPR lines, the 10-day-old seedlings were harvested. Three biological repeats were designed for each treatment.

To determine the salt tolerance of the transgenic lines, the 10-day-old *AtWRKY66* CRISPR lines treated with the same methods of seeds sterilization and germinating of seeds as mentioned above were placed in a 1/2 MS solid medium with 100 mM NaCl or 2.5 µM ABA; then, the taproots of each length of the WT and mutant plants were measured and analyzed two days after the treatments. For the treatments and experiments mentioned above, Col-0 and mutants were grown in a 1/2 MS solid medium under identically controlled conditions. Likewise, the seedlings were collected after the treatments with the same methods as described above and divided into two batches for the next analysis. The first was used for the physiological, biochemical and developmental analyses and the rest were used to extract the RNA for transcriptomic or expression analyses. All samples were rapidly frozen in liquid nitrogen and then stored at −80 °C. 

### 4.2. Genome-Wide Identification and Phylogenetic Analysis of the WRKY66 Homologs

The *AtWRKY66* information was searched from the public PLAZA (http://bioinformatics.psb.ugent.be/plaza/ (accessed on 13 September 2021)) [73]. Other protein sequences of the Group Ⅲ *AtWRKYs* were also downloaded from the PLAZA database, and were used as queries to perform BLASTP searches against 28 plant genomes with an E-value threshold of 1 × 10^−5^. The hit sequences with similarity to the query sequence were annotated with InterProScan (https://www.msi.umn.edu/sw/interproscan (accessed on 1 September 2020)) [74], and sequences containing “WRKY” were retained and used for the next analysis. All sequences were aligned by MAFFT v7.471 [75] and trimmed by trimAL v1.4.1 [76] with the thresholds of -gt = 0.9 and -cons = 20. The maximum-likelihood (ML) phylogenetic trees were reconstructed by using IQ-TREE [77] with the JTT+I+G+F model screened by ProtTest3 [78] with 2000 bootstraps.

### 4.3. Lineage-Specific Domains/Motifs and Molecular Evolution Analyses

The lineage-specific domains/motifs of the identified proteins were predicted by using the online MEME program [79] (https://meme-suite.org/meme/tools/meme (accessed on 9 October 2022)) with the parameter settings as follows: the occurrence rate of a single motif was no greater than one per sequence; the optimum motif width was between six and 50 amino acid residues; the maximum number of identified motifs was 25. Other parameters were default. The occurrences and relative positions of the motifs within the protein sequences were mapped to known conserved domains; others would be identified as lineage-specific or unique. 

To estimate the changes in the selection constraints in the duplicates found, the CodeML program implemented in the PAML v4.8 package [80] was used to detect the shifts in the substitution rates between specified foreground branches and background branches. Likelihood Ratio Tests (LRTs) were conducted to compare the likelihood of the branch-specific model: the one ratio model that assumed a constant ώ (dN/dS = nonsynonymous/synonymous substitutions) along tree branches (ώo), against a two-ratio model that assumed a different ratio for a designated subclade (foreground = ώf) relative to the remaining sequences (background = ώb). The clade C lineage was designated as a foreground branch. A chi-squared distribution was assumed for 2Δ*ℓ* with the difference between np2 and np1 as the freedom degree (difference between the parameter number of the one ratio and the two-ratio models) [81].

Multiple sequences were aligned using the online Clustal Omega tool (https://www.ebi.ac.uk/Tools/msa/clustalo/ (accessed on 1 October 2019)) with default settings. The alignment logos of the conserved domain were visualized by WebLogo (https://weblogo.threeplusone.com/ (accessed on 14 June 2004)). The pIs and Mws of the WRKY66s were calculated by using the online Compute pI/Mw tool (https://web.expasy.org/compute_pi/ (accessed on 1 March 1994)). The Ka and Ks values and their ratios of the aligned cDNA sequences of the candidate *WRKY66s* were calculated by using DNASP v5.10 [82]. The Jones–Taylor–Thornton (JTT) model in MEGA7.0 was used to calculate the genetic distances among the groups of WRKY66s and their overall mean distances [83]. 

### 4.4. Bioinformatic Analysis of the AtWRKY66 

The known stress-responsive *AtWRKYs* and their functions in stress response were also searched and retrieved in the PLAZA database. To dissect the distribution of the *cis*-acting elements, the upstream of the *AtWRKY66* promoters (the promoter region was only 389 bp because of the close distance between the promoter and upstream gene) was executed by the PLANTCARE (http://bioinformatics.psb.ugent.be/webtools/plantcare/html/ (accessed on 11 September 2000)). 

### 4.5. Gene Cloning, Vector Construction, Subcellular Localization Detection and Transcription Activity Assay of the AtWRKY66

The full-length cDNA of the *AtWRKY66* (*AT1G80590*) was amplified with a pair of specific primers, W66-T-F (KpnI) and W66-T-R (XbaI) (Appendix A), followed by being integrated into a T vector, yielding a plasmid 35S::*WRKY66*-T. The destination vectors, pCAMBIA1300-GFP and pGBKT7, were subsequently used to generate the constructs for subcellular localization detection and transcription regulation assay.

For subcellular localization detection, the coding sequence of the *AtWRKY66* without the stop codon from the vector 35S::*WRKY66*-T was amplified by using W66-GFP-F (KpnI) and W66-GFP-R (XbaI) primers (Appendix A), and cloned into a modified expression vector, pCAMBIA1300-GFP, at the KpnI/XbaI sites, forming a plasmid 35S::*WRKY66*-GFP, which was used for the transient expression in the epidermal cells of *N.benthamiana* leaves. The leaves agroinfiltrated with the 35S::*WRKY66*-GFP were cultured in a standard greenhouse and harvested at 48 h after *Agrobacterium tumefaciens* strain EHA105 infiltration [84]. The vectors, 35S::GFP (as control), were also transferred into the tobacco leaves in the same method as described above. The GFP fluorescence signals were excited by a 488-nm ion argon laser and observed by a Zeiss LSM 510 Meta confocal laser scanning microscope (Oberkochen, Germany) [85]. 

For the transcription activation assay, the full-length sequence of the *AtWRKY66* from the vector 35S::*WRKY66*-T was amplified by using specific primer pairs, pGBKT7-WRKY66-F (NdeI) and pGBKT7-WRKY66-R (EcoRI) (Appendix A), and introduced into the NdeI/EcoRI-digested pGBKT7 vector to yield the fusion vector pGBKT7-*WRKY66* (BD-WRKY66). The empty pGBKT7 vector was regarded as a negative control. All plasmids were transformed into the yeast strain AH109, respectively. The transformed yeast cells were streaked on SD/-Trp and SD/-Trp-His-Ade/X-α-Gal plates and the cell growth was observed at 30 °C for 2–3 days.

### 4.6. CRISPR/Cas9 Vector Construction, A. thaliana Transformation and the AtWRKY66 CRISPR Lines Generation 

The *pYAO*:hSpCas9-target-sgRNA fusion plasmid was used as a backbone in this study [86]. The sequence containing a specific restriction enzyme site (*NcoI*) upstream adjacent to the PAM as a gRNA candidate was selected for facilitating the subsequent identification of the positive plants. Subsequently, the oligonucleotide sequence of the cohesive terminals, ‘TGATT’ and ‘AAAC’, of the *BsaI* site were appended to the 5’ end as the sgRNA upper/lower primer, respectively (Appendix A). The annealed DNA was inserted into the *BsaI* digested *pYAO*:hSpCas9-target-sgRNA vector to generate the *pYAO*:hSpCas9-*WRKY66*-sgRNA which was transformed into the *A. tumefacien* strain EHA105 to infect the inflorescence of *Arabidopsis*. The transgenic plants were named as *WRKY66* knockdown. To screen the positive transgenic plants, the hygromycin resistant seedlings of the T0 generation plants were gathered to extract DNAs for the PCR using a pair of specific primers (Appendix A). The PCR products were digested by the *NcoI* and separated by 1% agarose gel electrophoresis. Among them, the products of the WT plants and those without the edited genome were cut, but those of the plants with the edited genome could not be cut because the enzyme sites were destroyed. The latter were selected and purified from the gel, and ligated to pGEM^®^-T Easy Vector Systems (Promega, Madison, WI, USA) for sequencing. The results were aligned by using Vector NTI Advanced v9.0 to show the detailed information of the mutations caused by the CRISPR/Cas9 system. The homozygous edited lines were selected for subsequent experiments.

### 4.7. RNA Extraction and Gene Expression Analysis

Total RNAs were extracted from the samples by using a Trizol reagent (Invitrogen, CA, USA) and cDNA libraries were synthesized by using PrimeScript RT Master Mix Perfect Real Time (Takara Co., Ltd., Beijing, China) according to the manufacturer’s instructions. qRT-PCR was performed on the Thermo Fisher Scientific StepOnePlus system with *Actin2* (*At3g18780*) as the control. The 10 µL reaction system for the PCR contained 5–50 ng of first-stand cDNA (4 µL), 5 pmol of each primer (0.4 µL), 5 μL SYBR Premix Ex Tap (2×) and 0.2 µL ROX. The program was as follows: initial activation at 95 °C for 5 min followed by 40 cycles of 95 °C for 30 s, and 60 °C for 34 s. Melting curves ranging from 60 °C to 95 °C followed by 0.5 °C/min were detected. Three separate biological replicates were performed with each treatment. The relative expression levels were calculated using the 2^−ddCT^ comparative method. The primers in this study were listed in Appendix A.

### 4.8. RNA-Seq, Deep Sequencing and Data Analysis

Total RNA (3 µg) was extracted separately from the seedling of the WT and *WRKY66* CRISPR lines with or without the 100 mM NaCl treatment for 48 h or the 10 µM ABA treatment for 12 h. The RNA integrity was evaluated by the Bioanalyzer 2100 system with the RNA Nano 6000 LabChip Kit (Agilent Technologies, Santa Clara, CA, USA) with RIN number >7.0. The RNA-seq libraries were constructed by using the whole transcriptome analysis kit for Illimina^®^ (NEB, Ipswich, MA, USA) with the manufacturer’s protocol. The RNA libraries were sequenced on the Illumina Hiseq™ platform to generate raw reads of 150 bp paired-end ones. The clean reads were obtained by eliminating the raw ones containing the adapters and ploy-N and the ones with low quality, followed by being mapped to the reference genome of *A. thaliana* by using TopHat v2.0.12. The quality control for mapping sequences to assembled transcripts was achieved with Bowtie v2.3.2 and the statistical power of RNA-seq data was calculated by the RNASeqPower Calculator [87,88]. The Transcripts per Million (TPM) represented the gene expression abundance [89]. The DEGs were screened by the EdgeR package, and the false discovery rate (FDR) adjusted *p*-value ≤ 0.05 and fold change ≥2 were used as the threshold [90,91]. The GOseq R package was used for the gene Ontology (GO) enrichment analysis of the DEGs, and the corrected *p*-value less than 0.05 was considered as a cut-off [91]. In addition, the identified DEGs were suffered from the Kyoto Encyclopedia of Genes and Genomes (KEGG) for identification of metabolic pathways. These raw data generated in this study have been deposited in the National Center for Biotechnology Information (NCBI) Short Read Archive (SRA) under BioProject ID PRJNA876747.

### 4.9. Germination Rate, Phenotypic, Physiobiochemical and Developmental Measurements

For germination assays, the *Arabidopsis* seeds were surface sterilized and then placed in the dark at 4 °C for 2 day before being sown on the 1/2 MS solid medium or the 1/2 MS solid medium supplied with 100 mM NaCl or 1 µM ABA. The plates were then incubated under a 16 h-light/8 h-dark light cycle at a constant temperature of 22 °C. Germination rates or greening rates were recorded after 10 day. For the biochemical analysis, the SOD, POD, CAT activities and REL were determined spectrophotometrically following previously described protocols [92]. Each sample contained at least 20 seedlings and each treatment underwent three biological replicates. DPS 7.05 Data Processing System Software was used for data analyses. For all analyses, the significance of differences was subjected to the Student’s *t*-test. An asterisk (*) indicated *p*-value < 0.05, and double asterisks (**) denoted *p* < 0.01. Sample variability was given as the standard deviation (SD) of the mean.

## 5. Conclusions

The *WRKY66* orthologue genes belonging to the Group Ⅲ *WRKYs* were angiosperm-specific, and had undergone motif loss events and relaxed purifying selection. The substitution rate tests displayed that the *WRKY66* lineage held significant differences compared with those of other lineages. The WRKY66 homologues had a relatively conserved WRKY and C2HC motifs with a high proportion of key amino acid residues in the average abundance. The *AtWRKY66* was a nuclear-localized transcription activator and strongly induced by the salt stress and ABA treatments. The knockdown plants of the *Atwrky66* generated by the CRISPR/Cas9 system showed lower SOD, POD and CAT activities and seed germination rate, and higher REL than those of the WT plants under both salt stress and ABA treatments, suggesting that the mutants increased the sensitivity to salt stress and ABA. Moreover, the RNA-seq analysis indicated that, under the two conditions, the expressions of the genes related to ABA signaling pathways and the stress-response of the *Atwrky66* knockdown plants showed similar trends. Importantly, several regulatory genes in the ABA signal transduction pathway were also significantly regulated. In conclusion, the *AtWRKY66* functions as a positive regulator in response to salt stress, in which an ABA-mediated signaling pathway may be involved.

## Figures and Tables

**Figure 1 ijms-24-03071-f001:**
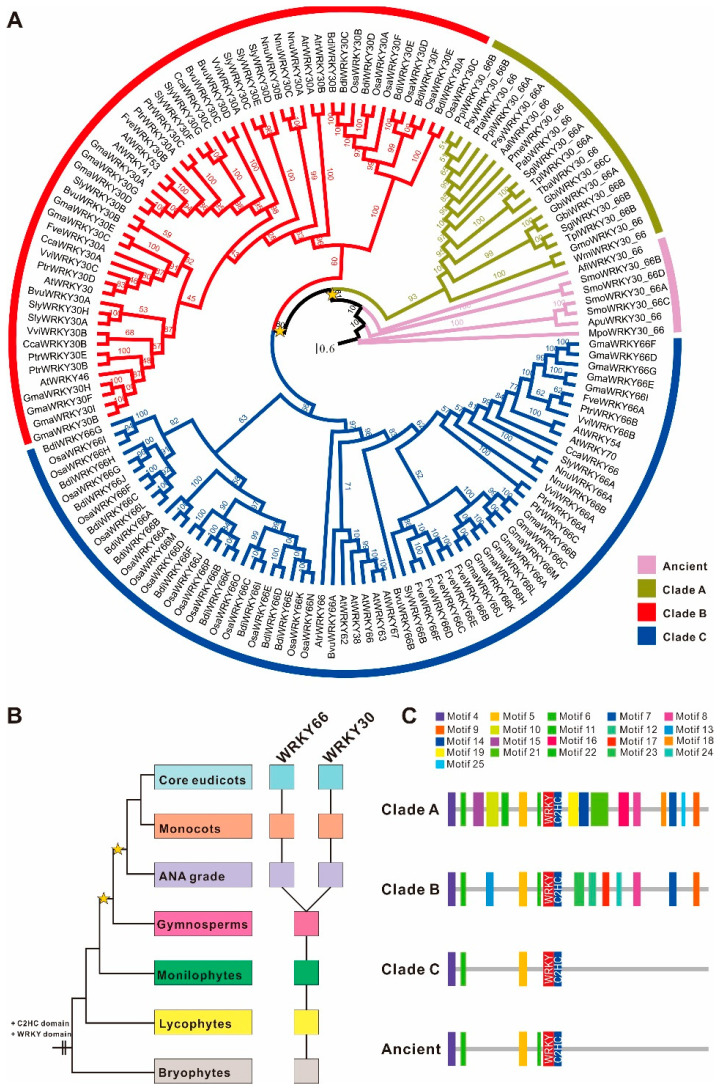
Phylogeny and domain architecture of the *WRKY66* homologs in land plants. (**A**) Phylogenetic tree of 145 *WRKY66* homologs identified from diverse land plants through maximum-likelihood analyses was conducted using the JTT+I+G+F model. The yellow star indicated the large-scale duplication events. Support values were shown for each node. The scale bar indicated the number of changes per site. Different gene lineages were marked with different colors. Duplication history (**B**) and domain architecture (**C**) of the *WRKY66* homologs. Filled squares indicate the presence of the corresponding members. The color of squares corresponds to the left organismal tree. The yellow stars in the tree indicated whole-genome duplication events. The diagram on the right demonstrated the domain/motif composition of different duplicates. The known conserved domains/motifs included: WRKY motif and C_2_HC-type zinc finger motif. The unnamed domains/motifs were lineage-specific and marked with the colors corresponding to the sequence logos in Appendix A.

**Figure 2 ijms-24-03071-f002:**
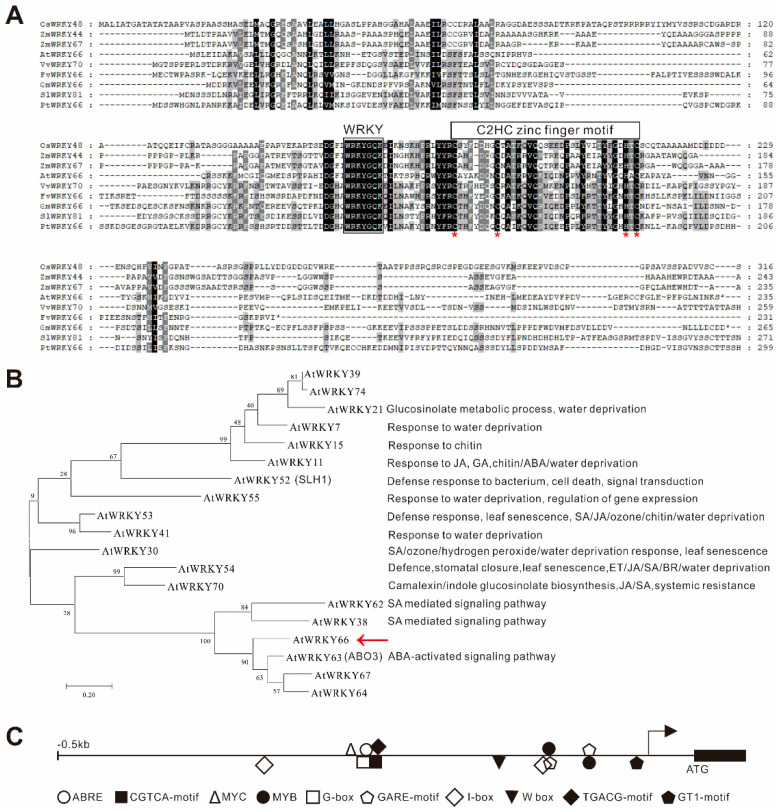
Characterization of the AtWRKY66 protein and gene. (**A**)**.** Multiple sequence alignment of AtWRKY66 with its homologs from *Oryza sativa* (Os05g0478400), *Zea mays* (Zm00001eb368850, Zm00001eb149550), *Vitis vinifera* (GSVIVG01032661001), *Solanum lycopersicum* (Solyc09g015770.3), *Glycine max* (Glyma.16G219800), *Fragaria vesca* (FvH4_6g09740) and *Populus trichocarpa* (Potri.016G137900). Identical and similar amino acids were shaded in black and light gray, respectively. The conserved WRKY motif and C2HC-type zinc-finger motif (C-X_7_-C-X_23_-H-X_1_-C) were marked by boxes and the residues of functional or structural importance were indicated by stars. (**B**). Phylogenetic analysis of *AtWRKY66* and WRKY proteins from *Arabidopsis*. IQ-TREE software with the maximum-likelihood (ML) method (2000 bootstrap repeats) was used to reconstruct the phylogenetic tree. Protein sequences used for the phylogenetic analysis were shown in Appendix A. Reported names and functions in the stress response of the known stress-responsive WRKY genes were given in parentheses in the tree and listed at right of the tree, respectively. The red arrow indicated the focus of this study. (**C**). Distribution of *cis*-elements in the promoter region of the *AtWRKY66* gene.

**Figure 3 ijms-24-03071-f003:**
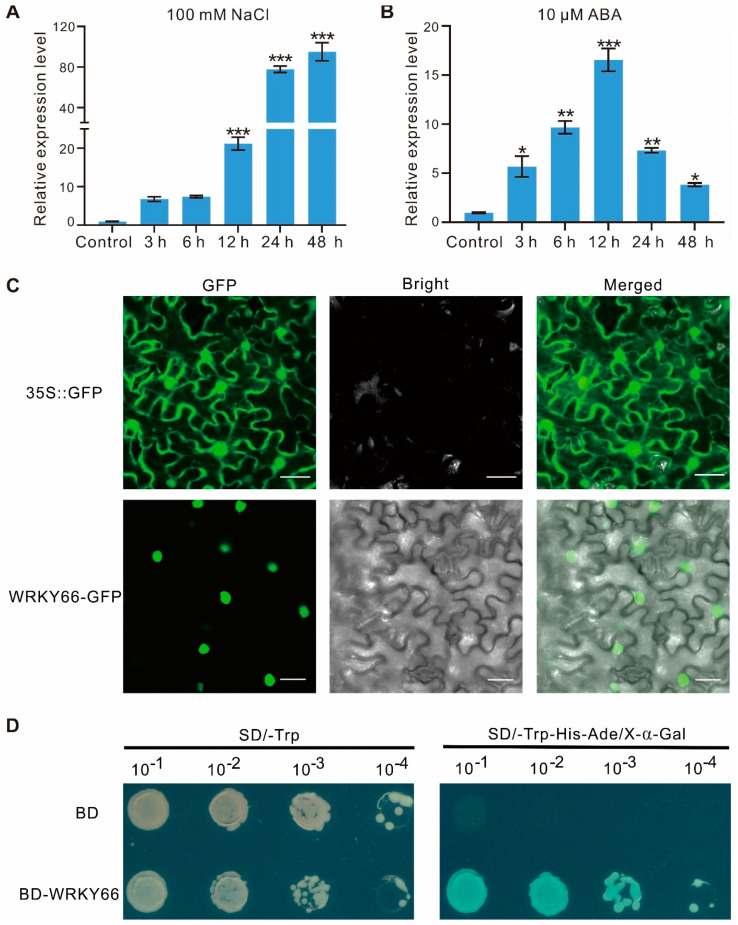
Expression characteristics, subcellular localization and transcription activity of *AtWRKY66*. The expression levels of *AtWRKY66* at different times under salt (**A**) and ABA (**B**) treatment. 10-day-old seedlings were treated with 100 mM NaCl or 10 µM ABA with the time course. After RNA extraction and reverse transcription, qRT-PCR assays were performed. *Actin* was used as an internal control for normalization. Error bars indicated the standard deviation (SD) on three replicates. Asterisks indicated a statistically significant difference compared with the control (* *p* < 0.05; ** *p* < 0.01; *** *p* < 0.001; Student’s *t*-test). (**C**). Subcellular localization of *AtWRKY66* in *N. benthamiana* leaves. *AtWRKY66*-GFP fusion proteins and GFP alone were each expressed transiently in leaves of *N. benthamiana* plants after 48 h post infiltration (hpi) and detected by confocal in the dark field for green fluorescence, white field for cell morphology and in combination, respectively. GFP, GFP fluorescence (green); Bright, bright-field; Merged, overlay of the GFP and bright-field images. Bar = 20 μm. (**D**). Identification of the transcription activation of *AtWRKY66*. Yeast cells carrying pGBKT7(BD)-*AtWRKY66* or a pBD empty vector (as a negative control) were streaked on SD/-Trp plates (top) or SD/-Trp-His-Ade plates supplemented with X-α-gal (bottom) for 3 day at 28 °C.

**Figure 4 ijms-24-03071-f004:**
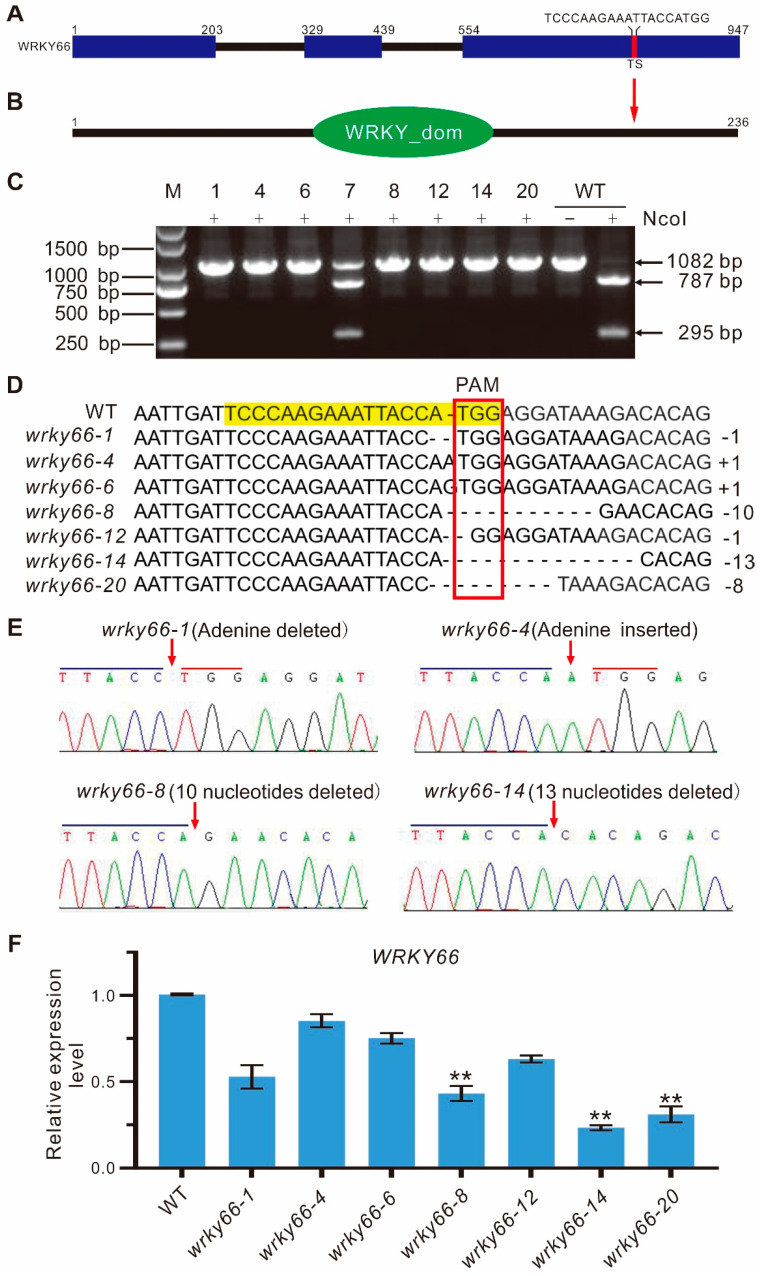
CRISPR/Cas9-mediated mutation details of the *AtWRKY66* gene in the T2 generation of transgenic plants. (**A**)**.** Schematic illustration of the target site (TS) in the *AtWRKY66* genomic sequence. The protospacer-adjacent motif (PAM) sequence was marked in red. The blue rectangle and black line indicated the exons and introns, respectively. (**B**). Location of TS in AtWRKY66 protein. The red arrow pointed to the TS site. (**C**). Identification of PCR products digested with restriction enzyme *NcoI* by gel electrophoresis. *NcoI*, PCR products digested with *NcoI*. The lengths of PCR products digested with *NcoI* were marked by arrow heads. The number on the lanes represents the different transgenic plant lines. The negative sign represented the no enzyme digestion, while the plus sign represented the enzyme digestion. M, DNA marker. (**D**). Comparisons and analyses of the PCR sequencing results of different transgenic plants. The gRNA sequence was labeled with light yellow and the PAM sequence was marked with a red box. (**E**). Detailed mutation information of the *AtWRKY66* transgenic line (T2). DNA fragments around the target sequences were amplified by PCR and then subjected to a sequencing analysis. The PAM and the sgRNA target sequences were indicated by blue and red lines, respectively. Red arrows indicated the insertion of an alanine (A) nucleotide or the position of deleted nucleotides. (**F**). Relative expression levels of *AtWRKY66* in mutants by qRT-PCR. The asterisk indicates that the fold changes of expression levels is greater < 0.5 than the WT plants (** *p* < 0.01).

**Figure 5 ijms-24-03071-f005:**
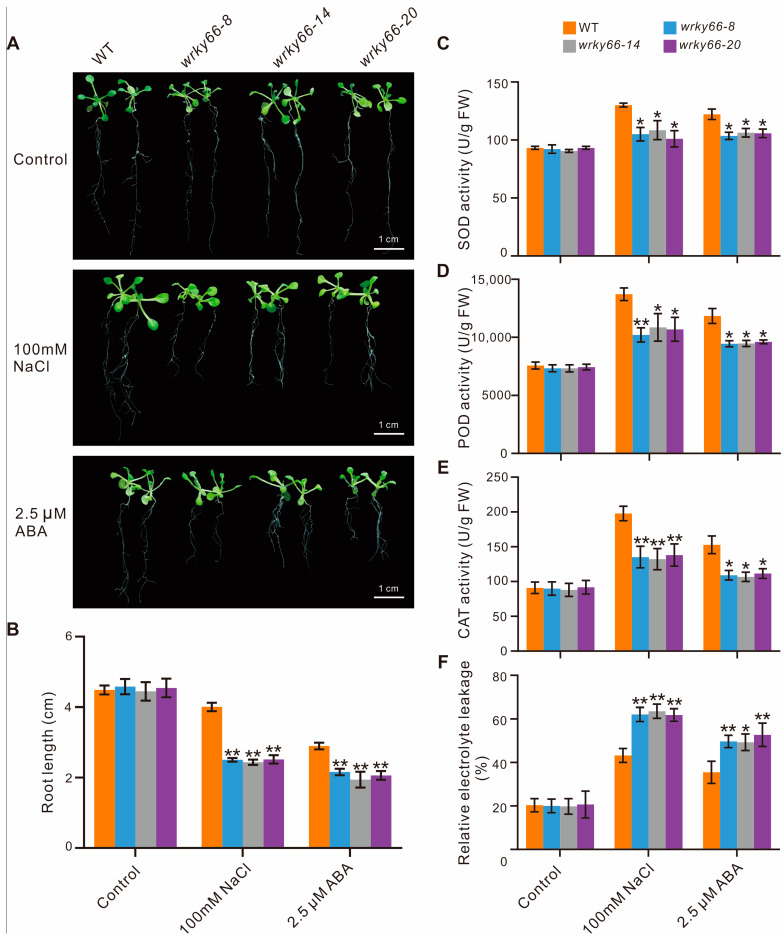
Decreased tolerance to salt stress and ABA in the *AtWRKY66* knockdown plants. (**A**). Phenotypic characteristics of the *wrky66* and WT seedlings under salt stress and ABA treatments. Growth of the *A. thaliana* plant on 1/2 MS solid medium (control) and 1/2 MS solid medium supplied with 100 mM NaCl or 2.5 µM ABA. (**B**). Root lengths were measured at eleven days after germination (i.e., after treatment with NaCl or ABA two days). (**C**–**F**). SOD, POD, CAT activity and REL of the mutant and WT seedlings under salt stress and ABA treatments, respectively. Control represented a normal growth condition. The error bars indicate the standard deviation from triplicate experiments. Asterisks indicated significant difference of the *wrky66* plants compared with the WT plants under the NaCl and ABA treatments (* *p* < 0.05; ** *p* < 0.01). WT, wild type; SOD, superoxide dismutase; POD, peroxidase; CAT, catalase; REL, Relative electrolyte leakage; FW, fresh weight.

**Figure 6 ijms-24-03071-f006:**
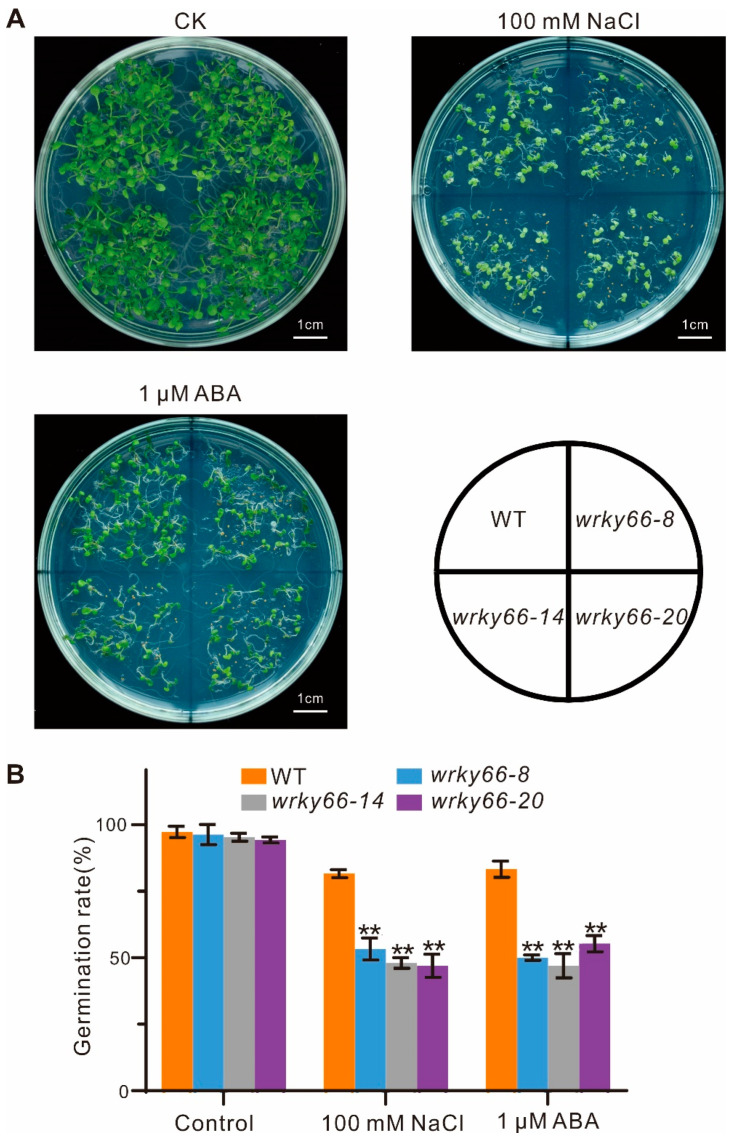
Effects of *AtWRKY66* mutations on seed germination under salt stress or ABA treatment. (**A**). Morphology of the germinating seeds of the *wrky66* plants compared with those of the WT plants under salt stress or ABA treatment. Pictures were taken on the 10th day after imbibition. Bar indicates 1 cm. (**B**). Seed germination was measured during the 10th day after imbibition in the WT and three *wrky66* mutants with or without treatment of NaCl (100 mM) or ABA (1 μM). Values were the mean ± SD, ** *p* < 0.01.

**Figure 7 ijms-24-03071-f007:**
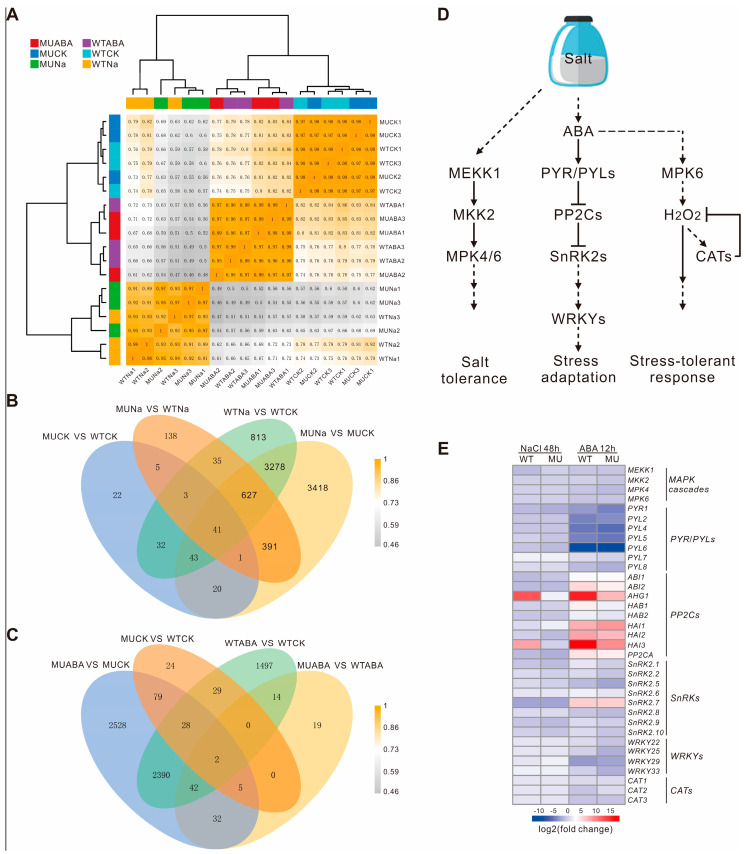
Changes of related gene expressions between the WT and *Atwrky66* transgenic plants after salt stress or ABA treatments. (**A**). Sample–sample correlation heatmap of 18 samples. (**B**). Venn diagrams of the differentially expressed genes (DEGs) in different comparisons after NaCl treatments. (**C**). Venn diagrams of the DEGs in different comparisons after ABA treatments. WT, wild type; MU, *wrky66-14*; Na, NaCl treatments; ABA, ABA treatments. (**D**). Schematic representation of the ABA-mediated signaling pathway based on KEGG analysis. *MPK*, mitogen-activated protein kinase; *MKK*, MPK kinase; *MEKK*, mitogen-activated protein kinase kinases; ABA, abscisic acid; *PYR*/*PYL*, PYRABACTIN RESISTANCE-LIKE; *PP2C*, protein phosphatase 2C; *SnRK2*, sucrose non-fermenting 1 (SNF1)-related protein kinase 2; *CAT*, catalase. (**E**). Heatmap of ABA-mediated signaling pathway related genes in the WT and *wrky66-14* plants under NaCl or ABA treatments. MU, *wrky66-14*; WT, wild type; NaCl, NaCl treatments; ABA, ABA treatments. The detailed information of the log_2_FC value was shown in Appendix A.

**Table 1 ijms-24-03071-t001:** Genetic distance between different groups of the *WRKY66s* calculated based on the amino acid sequences with the Jones–Taylor–Thornton (JTT) model.

	A	B	C
A			
B	0.601		
C	0.784	0.841	
Ancient	0.549	0.694	0.847

**Table 2 ijms-24-03071-t002:** Molecular evolutionary analysis of the *WRKY66s* using their whole cDNA sequences. N, number of sequences; Ka, the number of nonsynonymous substitutions per nonsynonymous site; Ks, the number of synonymous substitutions per synonymous site; ώ, Ka/Ks.

Clade	N	Ka	Ks	ώ	G + C Content
A	17	0.38965	0.55785	0.6985	0.456
B	54	0.40242	0.32688	1.2311	0.484
C	67	0.34505	0.72254	0.4776	0.501
Ancient	7	0.50928	0.47742	1.0667	0.576

**Table 3 ijms-24-03071-t003:** Likelihood ratio tests were performed for the *WRKY66* homologs. The *WRKY66* lineage was designated as foreground branch.

Family	Model	ώ (dN/dS)	lnL	2Δ*ℓ*	*df*	*p* Value
*WRKY66*	One Ratio	ώ0 = 0.37915	−124,561.161	26.467	1	2.68 × 10^−7^
	Two Ratios	ώf = 0.417126 ώb= 0.353674	−124,547.9273			

**Table 4 ijms-24-03071-t004:** Changes in the expression of the salt stress- and ABA- related genes in the WT and *wrky66* plants after the NaCl or ABA treatment.

	Gene Name	Gene ID	Salt Treatment 48 h/log_2_FC		ABA Treatment 12 h/log_2_FC	
			WT	*wrky66*	WT	*wrky66*
Salt stress-related genes	*AKR4C8*	AT2G37760	1.13	1.27	2.00	2.14
	*ANN1*	AT1G35720	1.13	1.56	1.51	1.88
	*BGLU24*	AT5G28510	3.59	1.85	6.34	5.72
	*GSTU17*	AT1G10370	1.20	1.25	1.83	2.42
	*SIGE*	AT5G24120	3.02	3.77	2.19	2.66
	*EGR1*	AT3G05640	1.40	1.75	4.51	4.74
	*LEA3*	AT1G02820	−1.22	−1.07	−1.90	−1.16
Salt stress- and ABA-related genes	*ALDH7B4*	AT1G54100	1.08	1.63	2.25	2.44
	*CHIC*	AT4G19810	2.26	3.21	1.07	1.95
	*NAC2*	AT5G39610	1.04	1.56	1.69	1.88
	*RD20*	AT2G33380	4.59	3.19	9.40	8.58
ABA-related genes	*ABA1*	AT5G67030	1.99	1.82	1.94	1.25
	*MYB102*	AT4G21440	5.07	4.78	10.26	9.17
	*PLDDELTA*	AT4G35790	1.04	1.58	1.71	1.79
	*SnRK2.6*	AT4G33950	1.86	1.82	2.26	1.51

## Data Availability

All the data generated in this study are included in this publishing article and its Appendix A. The transcriptome data has been deposited to the NCBI’s Short Read Archive (SRA) repository with the data set identifier PRJNA876747. Further inquiries can be directed to the corresponding author.

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
