# Peer review of "Evolution of the WRKY66 Gene Family and Its Mutations Generated by the CRISPR/Cas9 System Increase the Sensitivity to Salt Stress in Arabidopsis"

_ijms, 2023, doi:10.3390/ijms24043071_

Round 1
Reviewer 1 Report
The Manuscript titled “Evolution of WRKY66 gene family and its mutations generated by the CRISPR/Cas9 system increase the sensitivity to salt stress in Arabidopsis” focus on the evolution of WRKY66 gene and their characterization/ regulatory role in salt stress response.
Below some suggestions:
1- 1- Lines for review process should be numbered so that comments can be mentioned easily and accurately.
2- It is important to improve English.
3- There are many grammar and spelling mistakes eg misspelling of CRISPR/Cas-9. It would be best to have the MS checked by professional English writers such as the MDPI service or a native English speaker.
4- It would be better if the best/clearest image is selected for subcellular localization, especially the bright field of controlled (35S::GFP).

Author Response
The Manuscript titled “Evolution of WRKY66 gene family and its mutations generated by the CRISPR/Cas9 system increase the sensitivity to salt stress in Arabidopsis” focus on the evolution of WRKY66 gene and their characterization/ regulatory role in salt stress response.
Below some suggestions:
1- 1- Lines for review process should be numbered so that comments can be mentioned easily and accurately.
Answer Reviewer Q1: We firstly thank the reviewer for critical reading and for the helpful suggestions. As suggested, we have added the number of lines.
2- It is important to improve English.
Answer Reviewer Q2: Your help and suggestions are highly appreciated! The whole manuscript has been modified with English editing.
3- There are many grammar and spelling mistakes eg misspelling of CRISPR/Cas-9. It would be best to have the MS checked by professional English writers such as the MDPI service or a native English speaker.
Answer Reviewer Q3: We really appreciate your suggestion! We have carefully checked and corrected the grammar and spelling mistakes throughout the manuscript. In addition, the whole manuscript has been checked and revised by professional English writer.
4- It would be better if the best/clearest image is selected for subcellular localization, especially the bright field of controlled (35S::GFP).
Answer Reviewer Q4: We thank the reviewer for critical reading and for the helpful suggestions. I know that the result of the bright field of 35S::GFP could be selected the clearer image, unfortunately, I cannot find this origin data that may be missing in my exchanged jobs. In addition, the WRKY66-GFP fusion protein was obvious distributed in the nucleus (Fig. 3 in the revised manuscript). Thus, we think the current data also was good enough to indicate that they are subcellular in the nucleus. If necessary, please give me time to supplement the experiment.

Reviewer 2 Report
The manuscript IJMS-2173647 deals with very interesting and important topic of WRKY66 gene study in Arabidopsis mutant plants produced via CRISPR/Cas9 gene editing system in response to salinity stress and ABA treatment. Experiments and Results are good. Presentation is fine. Therefore, the manuscript can be conditionally accepted with a subject to minor but important revision as indicated below.
Major comments/corrections:
(1) This comment is more ‘major’ because is required more attention of authors to improve their section 4.1 in M&M (L441-461), where some important information was missed or confused. Therefore, several sub-points related to Major point 1 is listed as follows:
(1A) L445-446. “For expression profile analyses, 10-day-old seedlings were treated in 1/2 solid Murashige and Skoog (MS) medium…”. This phrase is absolutely confused. ’10-day-old seedling were treated’ How? Authors mentioned ‘solid MS media’ but it was not mentioned in any other cases with MS media (L453, L456, L592). Therefore, in all cases of used MS media, authors must add either ‘solid’ or ‘liquid’ media were used. Please add this information.
(1B) Next, authors used verbs that seedlings were ‘treated’ (L446), ‘placed’ (L453), ‘grown’ (L456), and ‘before sowing’ (L591). Only in the last case (L591), the description of seeds sterilization and sowing of seeds on [the surface of solid] MS media is relatively clear but this is unknown in other cases. If authors used in all cases exactly the same method of seeds sterilization and germinating of seeds on the surface of solid MS media, please describe it very clear in the first instance and make references to this description in all other spots. If authors used different methods, please either describe differences compared to the initial method or provide your full description in each case.
(1C) Next, it is still unclear to me how ’10-day-old seedling were treated’ (L445-446)? Where the seeds were germinated and seedlings were grown in the first 10 days (before the treatment)? Unclear... Please provide complete and clear description.
(1D) Next, how 100 mM NaCl and 10 uM ABA (L447) or 2.5 uM ABA (L453) were added to the media for the treatment of Arabidopsis seedlings? Does this mean that NaCl or ABA were added to MS media in the beginning, seeds were germinated and seedling were grown in such ‘supplemented’ media or not? Please provide clear description.
(1E) L448. “Samples were collected…”. What does mean ‘samples’ in the authors’ description? Is it only a single seedling or a part of individual seedling? Maybe this is a ‘bulk’ from several seedlings, as it was clear indicated in L598: “Each sample contained at least 20 seedlings…”. Please clarify number of seedlings used in bulks in each sample in other methods. L451. Similar clarification is required for number of biological replicates – three replicates means ‘three equal bulks or different?
Minor notes/errors:
(2) L33, L442, L624 and maybe in other spots. Please correct ‘WT’as ‘wild type’ but not ‘wide type’. The spelling and pronunciation of ‘wild’ and wide’ is very similar but the meaning is very different. Firstly, I thought that this is a simple mistyping but three times mistyping represent a systematic error. Please correct and use it in only in correct form. Interestingly, in L708, in the Legend of Figure 7, the term ‘wild type’ was used perfectly and correctly. Please use it everywhere in the manuscript.
(3) L35. The following phrase is incorrect “…salt and ABA stresses”. Salt (or salinity) is indeed abiotic stress but ABA (abscisic acid) is not a stress. Therefore, it is better to say ‘salt stress and ABA treatment’ here and in all similar cases.
(4) L38. The phrase ‘In brief…’ is unacceptable in the last sentence of Abstract. Please just delete it or replace for ‘Therefore,…’ or similar.
(5) L55. English has to be improved in the phrase: “…by activating the activity…”.
(6) L56. Please correct spelling of the kinase using a hyphen: ‘…non-fermenting…’.
(7) L193, L197, L263, L309, L377, L405, and in many other spots. Please correct unclear phrase “…ABA conditions”. It is much better to use ‘ABA treatment’ or ‘ABA treatments’ instead, as it was perfectly used in L388, 396, 399.
(8) L242-243. English has to be improved in the phrase for clear meaning: “…by pre-mature mature stop codons…”.
(9) L248. This is unclear, what authors want to say in the following phrase: “…on the mutation situations…”. Please modify for clear meaning.
(10) L273 and L277. Please correct unclear phrase “differential genes” and use ‘differentially expressed genes’ instead.
(11) L378. English has to be improved in the phrase: “…activation activity…”.
(12) L384. Authors must be more careful with phrase “water stress” because it can mean ‘not enough water – drought’ but also it can be ‘too much water – water-logging’. I suppose, authors want to say about drought, please correct.
(13) L398. The phrase about genes “…respond to salt tolerance…” is absurd. Probably, authors want to say ‘respond to salt stress’. Please correct.
(14) L410-412. In two sentences, the first one is finished with the reference “…signaling pathway in plants [6].” Therefore, the second sentence is the continuation of the previous one. It starts from the phrase: “In this study, the expression…”. In this fragment, the second sentence ‘In this study…’ means those indicated in reference [6], but this is wrong because the second sentence is ended with “(Figure 3A and 3B)” from the current manuscript. This is typical example of incorrect description and such negligence can confuse readers completely. Please correct the beginning of the second sentence as follows (as an example): ‘In contrast, in the current presented study…”.
(15) L563 and L565. Could you please explain and make your reference in the text of the manuscript why ROX was used in BioRad CFX96 qPCR instrument? I know that this is not requested at all, and I also checked the Manual of this equipment one more time. No ROX is required as a passive dye. Moreover, ROX cannot be used “as a passive reference standard to normalize the SYBR fluorescent signal” (L563-564). Normalization of SYBR amplification can be med using Reference (Housekeeping) gene. ROX can be only used to control evaporation in each well and this is typically used in qPCR instruments manufactured by Thermo Fisher Scientific but not by BioRad. Please clarify and make your corrections and add explanation in the text accordingly.
(16) L567. Please add your information if Melting curves were performed and analysed or not?
(17) L569. The sub-title is not exactly correct. I suggest changing it as follows: ‘RNA-seq, Deep Sequencing and Data Analysis’ or something similar because ‘deep sequencing’ was based on RNA-seq results.
(18) L600. Please correct ‘Student’s t-test’ starting from capitalized letter (not in regular case) because this is the name of a person, who described this method very long time ago. This is not related to the status like ‘student of College’, for example. Please use your correct spelling as it is present in L671, Legend of Figure 3.
(19) L604. “Strictly speaking…” is inappropriate start of Conclusion. Please either delete this phrase or replace it for more suitable.
(20) L629. Please either to add your Acknowledgments or delete this line completely.
(21) L654. Legend of Figure 1. Please remove repeats from the phrase: “…and C2HC zinc finger (C2HC) motif”.
(22) L678. Legend Figure 3. The term “mutation circumstances” is very strange and unclear. I am not sure at all what authors want to say. Maybe, this is just ‘mutation details’ or ‘details of mutations’. Please check and modify.
(23) L718. Supplementary Table S1. Please explain and insert it in the text or in the end of this Table Capture, why four names of Arabidopsis genes are indicated in red?
(24) L727. Supplementary Table S6. Please insert a column with the information about amplicon size, where this is possible.
Author Response
The manuscript IJMS-2173647 deals with very interesting and important topic of WRKY66 gene study in Arabidopsis mutant plants produced via CRISPR/Cas9 gene editing system in response to salinity stress and ABA treatment. Experiments and Results are good. Presentation is fine. Therefore, the manuscript can be conditionally accepted with a subject to minor but important revision as indicated below.
Major comments/corrections:
(1) This comment is more ‘major’ because is required more attention of authors to improve their section 4.1 in M&M (L441-461), where some important information was missed or confused. Therefore, several sub-points related to Major point 1 is listed as follows:
Answer Reviewer Q1: We firstly thank the reviewer for critical reading and for the helpful suggestions. As suggested, we have supplemented the important information was missed or confused in section 4.1 in Materials and Methods (Lines 437-445 in revised manuscript).
(1A) L445-446. “For expression profile analyses, 10-day-old seedlings were treated in 1/2 solid Murashige and Skoog (MS) medium…”. This phrase is absolutely confused. ’10-day-old seedling were treated’ How? Authors mentioned ‘solid MS media’ but it was not mentioned in any other cases with MS media (L453, L456, L592). Therefore, in all cases of used MS media, authors must add either ‘solid’ or ‘liquid’ media were used. Please add this information.
Answer Reviewer Q1A: Sorry for the confusion. We modified it as suggested (Lines 437-445 in revised manuscript).
(1B) Next, authors used verbs that seedlings were ‘treated’ (L446), ‘placed’ (L453), ‘grown’ (L456), and ‘before sowing’ (L591). Only in the last case (L591), the description of seeds sterilization and sowing of seeds on [the surface of solid] MS media is relatively clear but this is unknown in other cases. If authors used in all cases exactly the same method of seeds sterilization and germinating of seeds on the surface of solid MS media, please describe it very clear in the first instance and make references to this description in all other spots. If authors used different methods, please either describe differences compared to the initial method or provide your full description in each case.
Answer Reviewer Q1B: Again, apologies for the confusion caused. As suggested, we revised them.
(1C) Next, it is still unclear to me how ’10-day-old seedling were treated’ (L445-446)? Where the seeds were germinated and seedlings were grown in the first 10 days (before the treatment)? Unclear... Please provide complete and clear description.
Answer Reviewer Q1C: We sincerely appreciate all the valuable comments of the reviewer. Additional paragraphs have been added to the revised version to show relevant information.
“For the expression profile analyses, Arabidopsis seeds were surface sterilized and then placed in the dark at 4 °C for 2 days before sowing on 1/2 MS solid medium. The plates were then incubated under a 16 h-light/8 h-dark light cycle at a constant temperature of 22 °C. The 10-day-old seedlings were then transferred and treated with 100 mM NaCl (Sangon Biotech, Shanghai, China) or 10 µM ABA (Sangon Biotech, Shanghai, China) in 1/2 liquid Murashige and Skoog (MS) [72] medium (PhytoTechnology, USA) for 3 h, 6 h, 12 h, 24 h and 48 h, respectively.” (Lines 439-446 in revised manuscript)
(1D) Next, how 100 mM NaCl and 10 uM ABA (L447) or 2.5 uM ABA (L453) were added to the media for the treatment of Arabidopsis seedlings? Does this mean that NaCl or ABA were added to MS media in the beginning, seeds were germinated and seedling were grown in such ‘supplemented’ media or not? Please provide clear description.
Answer Reviewer Q1D: Sorry for the confusion. The Arabidopsis seeds were surface sterilized and then placed in the dark at 4 °C for 2 days before sowing on 1/2 MS solid medium for 10 days. The plates were then incubated under a 16 h light/8 h dark–light cycle at a constant temperature of 22 °C. Then, 10-day-old seedlings were transferred and treated with 100 mM NaCl, 10 µM ABA or 2.5 µM ABA in 1/2 liquid MS medium for corresponding experiments.
(1E) L448. “Samples were collected…”. What does mean ‘samples’ in the authors’ description? Is it only a single seedling or a part of individual seedling? Maybe this is a ‘bulk’ from several seedlings, as it was clear indicated in L598: “Each sample contained at least 20 seedlings…”. Please clarify number of seedlings used in bulks in each sample in other methods. L451. Similar clarification is required for number of biological replicates – three replicates means ‘three equal bulks or different?
Answer Reviewer Q1E: We appreciate greatly for this excellent suggestion. We have modified them as suggested in the revised version.
Minor notes/errors:
(2) L33, L442, L624 and maybe in other spots. Please correct ‘WT’as ‘wild type’ but not ‘wide type’. The spelling and pronunciation of ‘wild’ and wide’ is very similar but the meaning is very different. Firstly, I thought that this is a simple mistyping but three times mistyping represent a systematic error. Please correct and use it in only in correct form. Interestingly, in L708, in the Legend of Figure 7, the term ‘wild type’ was used perfectly and correctly. Please use it everywhere in the manuscript.
Answer Reviewer Q2: Sorry for the mistake. We have carefully checked and corrected throughout the manuscript.
(3) L35. The following phrase is incorrect “…salt and ABA stresses”. Salt (or salinity) is indeed abiotic stress but ABA (abscisic acid) is not a stress. Therefore, it is better to say ‘salt stress and ABA treatment’ here and in all similar cases.
Answer Reviewer Q3: As suggested by the reviewer, we have corrected “salt and ABA stresses” into “salt stress and ABA treatment”.
(4) L38. The phrase ‘In brief…’ is unacceptable in the last sentence of Abstract. Please just delete it or replace for ‘Therefore,…’ or similar.
Answer Reviewer Q4: As suggested, we changed “In brief” into “Therefore”.
(5) L55. English has to be improved in the phrase: “…by activating the activity…”.
Answer Reviewer Q5: Sorry for the confusion, we have improved the expression.
(6) L56. Please correct spelling of the kinase using a hyphen: ‘…non-fermenting…’.
Answer Reviewer Q6: As suggested, we corrected it.
(7) L193, L197, L263, L309, L377, L405, and in many other spots. Please correct unclear phrase “…ABA conditions”. It is much better to use ‘ABA treatment’ or ‘ABA treatments’ instead, as it was perfectly used in L388, 396, 399.
Answer Reviewer Q7: Thank you so much! We have done as suggested by reviewer.
(8) L242-243. English has to be improved in the phrase for clear meaning: “…by pre-mature mature stop codons…”.
Answer Reviewer Q8: Your suggestion is highly appreciated! We have improved the sentence as suggested.
(9) L248. This is unclear, what authors want to say in the following phrase: “…on the mutation situations…”. Please modify for clear meaning.
Answer Reviewer Q9: As suggested, we modified the sentence: “…based on its mutation situations and expression levels in CRISPR lines…”.
(10) L273 and L277. Please correct unclear phrase “differential genes” and use ‘differentially expressed genes’ instead.
Answer Reviewer Q10: Again, sorry for the mistake. We have carefully checked and corrected throughout the manuscript.
(11) L378. English has to be improved in the phrase: “…activation activity…”.
Answer Reviewer Q11: As suggested, we changed “…activation activity…” into “…transcription activation function…”.
(12) L384. Authors must be more careful with phrase “water stress” because it can mean ‘not enough water – drought’ but also it can be ‘too much water – water-logging’. I suppose, authors want to say about drought, please correct.
Answer Reviewer Q12: As suggested, we have corrected it.
(13) L398. The phrase about genes “…respond to salt tolerance…” is absurd. Probably, authors want to say ‘respond to salt stress’. Please correct.
Answer Reviewer Q13: Sorry for the mistake, we have corrected it.
(14) L410-412. In two sentences, the first one is finished with the reference “…signaling pathway in plants [6].” Therefore, the second sentence is the continuation of the previous one. It starts from the phrase: “In this study, the expression…”. In this fragment, the second sentence ‘In this study…’ means those indicated in reference [6], but this is wrong because the second sentence is ended with “(Figure 3A and 3B)” from the current manuscript. This is typical example of incorrect description and such negligence can confuse readers completely. Please correct the beginning of the second sentence as follows (as an example): ‘In contrast, in the current presented study…”.
Answer Reviewer Q14: We apologize for the ambiguity. We revised into “Likewise, in the current presented study…”.
(15) L563 and L565. Could you please explain and make your reference in the text of the manuscript why ROX was used in BioRad CFX96 qPCR instrument? I know that this is not requested at all, and I also checked the Manual of this equipment one more time. No ROX is required as a passive dye. Moreover, ROX cannot be used “as a passive reference standard to normalize the SYBR fluorescent signal” (L563-564). Normalization of SYBR amplification can be med using Reference (Housekeeping) gene. ROX can be only used to control evaporation in each well and this is typically used in qPCR instruments manufactured by Thermo Fisher Scientific but not by BioRad. Please clarify and make your corrections and add explanation in the text accordingly.
Answer Reviewer Q15: Sorry for the mistake, we have carefully checked and corrected the statement about the expression experiment again. The qRT-PCR was used in Thermo Fisher Scientific StepOnePlus system, instead of the BioRad CFX96 qPCR instrument.
(16) L567. Please add your information if Melting curves were performed and analysed or not?
Answer Reviewer Q16: We also performed and analyzed the melting curves.
(17) L569. The sub-title is not exactly correct. I suggest changing it as follows: ‘RNA-seq, Deep Sequencing and Data Analysis’ or something similar because ‘deep sequencing’ was based on RNA-seq results.
Answer Reviewer Q17: We appreciate the reviewer’s suggestion. We modified the sub-title “RNA-seq, deep sequencing and data analysis” as suggested by reviewer.
(18) L600. Please correct ‘Student’s t-test’ starting from capitalized letter (not in regular case) because this is the name of a person, who described this method very long time ago. This is not related to the status like ‘student of College’, for example. Please use your correct spelling as it is present in L671, Legend of Figure 3.
Answer Reviewer Q18: Thanks for pointing this out, we have corrected it.
(19) L604. “Strictly speaking…” is inappropriate start of Conclusion. Please either delete this phrase or replace it for more suitable.
Answer Reviewer Q19: We agree that this content did not fit very well here. We have deleted.
(20) L629. Please either to add your Acknowledgments or delete this line completely.
Answer Reviewer Q20: Thanks for pointing this out, we have deleted it.
(21) L654. Legend of Figure 1. Please remove repeats from the phrase: “…and C2HC zinc finger (C2HC) motif”.
Answer Reviewer Q21: As suggested, we modified it.
(22) L678. Legend Figure 3. The term “mutation circumstances” is very strange and unclear. I am not sure at all what authors want to say. Maybe, this is just ‘mutation details’ or ‘details of mutations’. Please check and modify.
Answer Reviewer Q22: Thanks for pointing this out, we have revised it.
(23) L718. Supplementary Table S1. Please explain and insert it in the text or in the end of this Table Capture, why four names of Arabidopsis genes are indicated in red?
Answer Reviewer Q23: Sorry for the confusion! That is our mistake. We changed the red of four names of Arabidopsis genes into black same as others.
(24) L727. Supplementary Table S6. Please insert a column with the information about amplicon size, where this is possible.
Answer Reviewer Q24: We appreciate the reviewer’s nice comments and suggestion. The column with the information about amplicon size has added as suggested.

Reviewer 3 Report
In this study, the authors evaluated the relationship between WRKY66 gene family and its mutations generated by the CRISPR/Cas9 system and the sensitivity to salt stress in Arabidopsis.
This is a very interesting study, because salt stress affects the growth and development of plants throughout their life cycle, with serious consequences on productivity.
The reviewed paper is of high quality and represents the results of completely finished research having a clear practical value. Although it is not a new topic, it can contribute to improving knowledge on the nuclear protein, inducible transcription activator signaling pathway and stress-response related genes. The work is valuable but requires a few corrections before being published.
Authors should clearly define the purpose of their research. In the last paragraph of Introduction section, they write what has been done at work, which looks more like conclusions than the purpose of the paper.
Some of the sentences are difficult to read, such as L85-92. Please check the whole manuscript and make these sentences friendly to read.
Please write in italics the scientific name in reference no. 85.
It is interesting to know if the authors propose to continue and deepen this study.
Author Response
In this study, the authors evaluated the relationship between WRKY66 gene family and its mutations generated by the CRISPR/Cas9 system and the sensitivity to salt stress in Arabidopsis.
This is a very interesting study, because salt stress affects the growth and development of plants throughout their life cycle, with serious consequences on productivity.
The reviewed paper is of high quality and represents the results of completely finished research having a clear practical value. Although it is not a new topic, it can contribute to improving knowledge on the nuclear protein, inducible transcription activator signaling pathway and stress-response related genes. The work is valuable but requires a few corrections before being published.
Authors should clearly define the purpose of their research. In the last paragraph of Introduction section, they write what has been done at work, which looks more like conclusions than the purpose of the paper.
Answer Reviewer Q1: We are sorry for not giving a clear introduction to the purpose of the paper. We have modified the paragraph to address this issue.
“Here, we investigated WRKY66 homologs using bioinformatics methods and analyzed the taxonomic distribution, structural diversification following duplication, molecular evolution and phylogenetic relationships throughout plant lineages. To better understand the function of Arabidopsis WRKY66 gene (AtWRKY66), expression patterns was surveyed under salt stress and ABA treatment. Moreover, the subcellular localization analysis and transcription activation assay of AtWRKY66 were also performed and observed. Furthermore, the CRISPR lines of AtWRKY66 generated by CRISPR/Cas9 system was performed and analyzed the phenotypes under salt stress and ABA treatment including the relative electrolyte leakage (REL) intensity, seed germination rate, and the activities of superoxide dismutase (SOD), peroxidase (POD) and catalase (CAT). Finally, RNA-seq and quantitative real-time PCR (qRT-PCR) analyses of WRKY66 CRISPR lines under salt stress and ABA treatment were performed. Overall, these results provided a new perspective on the origin and evolutionary history, and laid a foundation for further functional research of plant WRKY66s.”
Some of the sentences are difficult to read, such as L85-92. Please check the whole manuscript and make these sentences friendly to read.
Answer Reviewer Q2: Sorry for the confusion, the following paragraphs have been improved to address the reviewer’ concern:
“It is well-known that the loss-of-function mutant is the crucial component in gene function study and, in recent years, genome editing has been an important dynamic technique to acquire the mutant. In gene editing, several sequence-specific nucleases, e.g., zinc-finger nucleases (ZFNs), transcription activator-like effector nucleases (TALENs), and clustered regularly interspaced short palindromic repeats (CRISPR)-associated protein 9 (Cas9) systems have been successfully applied in plant gene-editing [27]. Among them, the CRISPR/Cas9 system has been widely employed to edit plant genomes due to its high-efficiency, stabilization and simplicity [28], this system enables the generation of specific double-stranded DNA breaks (DSBs) at a site complementary to the guide RNA (gRNA) sequence 2-4 bp upstream of the protospacer adjacent motif (PAM) sequence [29], which allows this system to efficiently implement point mutations including the nucleotide insertions or deletions of target genes”.
In addition, the whole manuscript has been revised with English editing.
Please write in italics the scientific name in reference no. 85.
Answer Reviewer Q3: As suggested, we have corrected it.
It is interesting to know if the authors propose to continue and deepen this study.
